# TFS-NeRF: Template-Free NeRF for Semantic 3D Reconstruction of Dynamic Scene

**Sandika Biswas**[1,2], **Qianyi Wu**[1], **Biplab Banerjee**[2], and **Hamid Rezatofighi**[1]

[1]Faculty of IT, Monash University
[2]Indian Institute of Technology (IIT), Bombay

## Abstract

Despite advancements in Neural Implicit models for 3D surface reconstruction, handling dynamic environments with interactions between arbitrary rigid, non-rigid, or deformable entities remains challenging. The generic reconstruction methods adaptable to such dynamic scenes often require additional inputs like depth or optical flow or rely on pre-trained image features for reasonable outcomes. These methods typically use latent codes to capture frame-by-frame deformations. Another set of dynamic scene reconstruction methods, are entity-specific, mostly focusing on humans, and relies on template models. In contrast, some template-free methods bypass these requirements and adopt traditional LBS (Linear Blend Skinning) weights for a detailed representation of deformable object motions, although they involve complex optimizations leading to lengthy training times. To this end, as a remedy, this paper introduces TFS-NeRF, a template-free 3D semantic NeRF for dynamic scenes captured from sparse or single-view RGB videos, featuring interactions among two entities and more time-efficient than other LBS-based approaches. Our framework uses an Invertible Neural Network (INN) for LBS prediction, simplifying the training process. By disentangling the motions of interacting entities and optimizing per-entity skinning weights, our method efficiently generates accurate, semantically separable geometries. Extensive experiments demonstrate that our approach produces high-quality reconstructions of both deformable and non-deformable objects in complex interactions, with improved training efficiency compared to existing methods. The code and models will be available on our *github page*.

## 1 Introduction

With the rapid advancements in deep learning, the field of 3D geometry reconstruction of static and dynamic scenes and objects has experienced significant transformation, largely due to its crucial role in diverse applications such as Augmented Reality (AR), Virtual Reality (VR), robotics, autonomous navigation systems, and human-robot interactions (HRI). While primarily focused on novel view synthesis, Neural Implicit models (NeRFs) [1] have recently made remarkable strides in 3D surface reconstruction due to their ability to learn detailed geometry without needing prior knowledge about the shape of the scene elements or direct 3D supervisions [2, 3, 4, 5, 6]. Despite these advancements, challenges remain in developing models that effectively generalize across varied real-world dynamic environments, particularly those involving *arbitrary rigid, non-rigid, or deformable entities* (such as humans or animals) engaged in complex *interactions*. Moreover, achieving *semantic reconstruction* that accurately captures the geometry and positioning of each semantic scene element independently is essential for enhancing functionalities in applications like AR, VR, and HRI.

38th Conference on Neural Information Processing Systems (NeurIPS 2024).

In NeRF-based dynamic scene reconstruction, the focus has predominantly been on human reconstruction [7, 8, 9, 10], utilizing template models such as SMPL [11], and CAPE [12]. However, these methods struggle with generalizability to arbitrary deformable entities and primarily concentrate on single-entity reconstructions, neglecting interactions among multiple entities, such as interactions of humans with scene objects. Notably, HOSNeRF [13] by Liu *et al*. addresses human-object interactions but remains dependent on the SMPL model.

Concurrently, some dynamic NeRFs [14, 15, 16, 17, 18, 19, 20, 21, 22, 23, 24, 25, 26] aim beyond human and focus on either rendering or geometry reconstruction of generic scene/object. The surface reconstruction methods mostly require additional data, such as depth or optical flow, to optimize the underlying 3D. For instance, [16, 17] focus on building shape-prior-free 3D models for deformable entities. However, besides requiring a large number of casual videos with diverse view coverage of the subject, these approaches heavily depend on high-quality optical flow for supervision, which makes their reconstruction quality subject to the accuracy of the pre-trained models for generating such data. Moreover, generally, the dynamic NeRF methods use latent codes to learn per-frame deformations [15, 27], which may not be sufficient for capturing accurate articulation of arbitrary deformable objects.

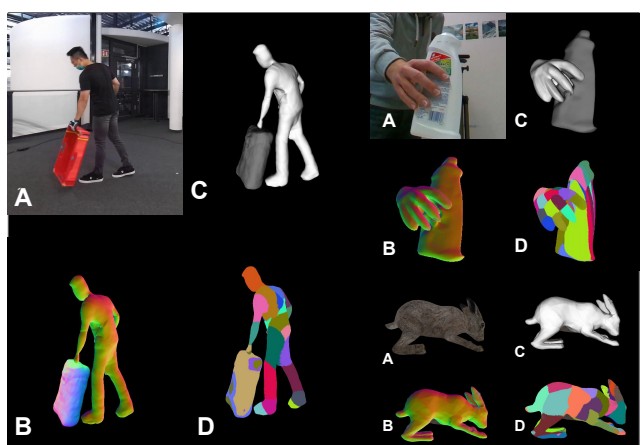

Figure 1: Existing dynamic-NeRF models struggle to generate plausible 3D reconstructions for generic dynamic scenes featuring humans and objects engaged in complex interactions. In this work, we introduce a Neural Radiance Field model designed for 3D reconstruction of such generic scenes, captured using a sparse/single-view video, capable of producing plausible geometry for each semantic element within the scene. In this figure, A: Input RGB, B: predicted normal map, C: predicted semantic reconstruction, and D: predicted skinning weight.

In a concurrent work, Li *et al*. [28] bypass the need for such additional information and develop a template-free model from only sparse view RGB videos. They also learn a more accurate representation of body deformation, the forward LBS [11] (*i.e*., mapping of canonical points to posed or view space), which helps them achieve impressive shape reconstruction results for different deformable entities. However, their method suffers from a lengthy training convergence time. Furthermore, none of these template-free approaches have focused on interactions between more than one entity nor achieved semantic reconstruction of scenes, highlighting a gap in the current methodologies.

**Our contributions:** In this paper, we aim to develop a time-efficient 3D semantic NeRF for dynamic scenes captured from sparse view RGB videos involving interactions between two rigid, non-rigid, or deformable objects. Built upon [28], we propose a novel framework to learn the forward LBS by utilizing INN [29] to bypass the computationally demanding root-finding approach used in [28]. This adjustment boosts the efficiency of our training process, which is shown with an experiment in the results section. Additionally, our proposed framework can produce semantically separable reconstructions for the scene entities. The challenge of building template-free 3D models for two dynamic entities engaged in complex interactions is amplified by occlusions and their diverse shapes or deformations. To tackle these challenges, we propose a strategy to disentangle the motions of distinct entities within the scene. Specifically, we first perform semantic-aware ray sampling and learn the independent transformation of each entity from the deformed space to the canonical space to optimize per-entity skinning weight. This process allows us to accurately generate the individual Signed Distance Fields (SDF) from the disentangled canonical points, enhancing our model's ability to handle complex dynamic environments under interactions effectively. The efficacy of our proposed method is demonstrated through comprehensive experiments conducted on several datasets featuring human-object, hand-object interactions, and animal movements. Results consistently show superior performance compared to the best-performing state-of-the-art methods. Our noteworthy technical contributions are, therefore:

- We introduce a time-efficient template-free NeRF-based 3D reconstruction method for dynamic

scenes from sparse multi-view/single videos featuring two interacting entities.
- Our approach extends to the semantic reconstruction of dynamic scenes, emphasizing the detailed capture of each entity's explicit geometry within the scene.
- The efficacy of our proposed method is demonstrated through comprehensive experiments conducted on diverse datasets featuring interaction between rigid, and non-rigid entities.

## 2   Related Works

**Human-object reconstruction:** Several model-based approaches [30, 31, 32, 33, 34, 35, 36] have explored 3D reconstruction of humans and objects under interactions. However, a common limitation is their reliance on parametric models (*e.g.*, SMPL) for human reconstruction. While SMPL provides a robust base, it limits flexibility in reconstructing diverse deformable entities and cannot generalize beyond humans. Also, it does not capture finer details such as hair dynamics and clothing deformations. In contrast, NeRF-based reconstruction offers a promising alternative for capturing detailed geometric information but mainly focuses on human reconstruction under a dynamic environment. Yet, only a few NeRF-based methods [13, 37, 38] have addressed the reconstruction involving multiple objects. For example, Jiang *et al.* [37] consider modeling the background, along with dynamic human by designing two separate NeRFs *i.e.*, *human NeRF* and *background NeRF*. In a recent work, HOSNeRF, Liu *et al.* [13] introduce a NeRF-based approach for free-viewpoint rendering of dynamic scenes featuring human-object interactions. *Even using the NeRF-based approach, these methods still rely on the SMPL model for initialization. Moreover, unlike our method, most of these methods focus on novel view or pose synthesis and do not emphasize detailed geometry reconstruction.*

**Generic Scene Reconstruction:** Some NeRF-based methodologies [14, 15, 27, 40, 39, 41, 25, 26] offer generic scene reconstruction under a dynamic environment without relying on any parametric template models, making them favorable for extending to any deformable object reconstruction. Pumarola *et al.* [14] extend the traditional NeRF framework by integrating time as an additional input, which facilitates the mapping of each frame's deformed scene to a canonical space. Building on this, T-NeRF [42] utilizes time-

| Methods | Template-free | No pre-trained features | Reconstructs |
|---|---|---|---|
| Vid2Avatar [39] | ✗ | ✓ | single entity |
| AnimatableNeRF [7] | ✗ | ✓ | single entity |
| SDF-PDF [8] | ✗ | ✓ | single entity |
| HumanNeRF [40] | ✗ | ✓ | single entity |
| HOSNeRF [13] | ✗ | ✓ | multiple entities |
| NDR [15] | ✓ | ✓ | single entity |
| HyperNeRF [27] | ✓ | ✓ | single entity |
| D-NeRF [14] | ✓ | ✓ | single entity |
| BANMO [16] | ✓ | ✗ | single entity |
| RAC [17] | ✓ | ✗ | single entity |
| TAVA [28] | ✓ | ✓ | single entity |
| **Ours** | ✓ | ✓ | **multiple entities *with semantic*** |

Table 1: Our approach vs. existing dynamic NeRFs.

varying latent codes to condition the NeRF, improving training speed and rendering quality. Cai *et al.* [15] propose using an INN to learn the mapping between deformed and canonical space and show impressive surface reconstruction results optimized from RGB-D videos. HyperNeRF [27] incorporates an additional Multilayer Perceptron to learn the frame-specific topology variations via ambient codes, capturing scene deformations more effectively. However, these methods mainly rely on latent codes to capture the relation between frames at different timestamps or topological variations within a frame. A recent approach Tensor4D [25] represents dynamic scenes as a 4D spatiotemporal tensor but does not account for the relative motions between scene elements. *In contrast, our approach considers learning the structural relations between different parts of the scene or the object under reconstruction* e.g.*, body parts of humans in case of human reconstruction,* etc.

**Reconstruction with predicted LBS:** Several methods have recently emerged that aim to develop generic NeRFs focusing on arbitrary deformable object reconstruction [16, 17, 28]. These methods utilize more specific topological representation, *i.e.*, LBS, that helps understand how different parts of the deformable body are connected and how they deform from the canonical pose under motion. Specifically, given the forward LBS weight $\mathbf{w}(\mathbf{x_c})$ for a canonical point $\mathbf{x_c}$ on the surface, the corresponding deformed point $\mathbf{x_v}$ (in the viewing space, *i.e.*, per-frame observations) is computed using a weighted combination of bone transformation matrices $\mathbf{B}$ [11] defined as,

$$\mathbf{x_v} = LBS(\mathbf{w}(\mathbf{x_c}), \mathbf{B}, \mathbf{x_c}) = \sum_{b=1}^{n_b} \mathbf{w}(\mathbf{x_c}).B_b.\mathbf{x_c} \tag{1}$$

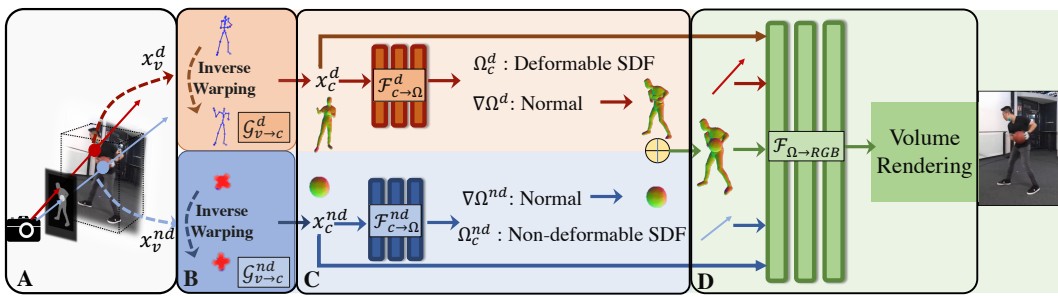

Figure 2: **Overview of the system. A:** To produce a semantically separable reconstruction of each element, first, we perform a semantic-aware ray sampling. Given a 2D semantic segmentation mask, we shoot two sets of rays and sample two sets of 3D points for differentiating the deformable and non-deformable entities of the scene, $\{x_v^d\}_{i=1}^N$, $\{x_v^{nd}\}_{i=1}^N$ under interactions. **B:** Next, each set of points is transformed from the deformed/view space (input frame) to its respective canonical space by inverse warping enabled by the learned forward LBS (Details are presented in Fig. 3. **C:** Then the individual geometry is predicted at the canonical space in the form of canonical SDFs by two independent SDF prediction networks $\mathcal{F}_{c->\Omega}^j(\theta)$ for the deformable and non-deformable entities denoted as $j \in \{d, nd\}$. **D:** Finally, the output SDFs are used to predict a composite scene rendering. Both these branches are optimized jointly using the RGB reconstruction loss.

where $n_b$ represents the total number of bones. Unlike human reconstruction NeRFs [7, 8, 9, 10], these methods learn this skinning weight function $\mathbf{w}(\mathbf{x_c})$, that allows for the flexible adaptation of these models to any deformable objects. Yang *et al.* [16] learns a forward-backward deformation field for mapping between canonical and deformed space and represents the skinning weight using a set of Gaussians defined around the bones. Whereas Li *et al.* [28] and Chen *et al.* [41] learn the skinning weights and formulate the mapping from deformed to canonical spaces as an iterative root-finding problem. They solve this problem using Broyden's method, which involves matrix-vector multiplications and computationally expensive matrix inversions at each iteration, leading to a lengthy training convergence time. *Additionally, these methods consider reconstructing only a single entity within the scene. In contrast, our method addresses the reconstruction of two moving objects under complex interactions, which is more challenging due to the highly unconstrained nature of the problem.*

## 3 Methodology

**Overview.** Given sparse-/single-view videos of deformable objects, such as hands, humans, or animals, interacting with non-rigid/rigid objects like balls or boxes, our goal is to accurately learn the semantically separable geometry of each entity under interactions. Fig. 2 provides an illustrative overview of our methodology, highlighting the following key components and steps involved in the process. A) *Semantic-aware ray casting and sampling* to distinguish individual elements within the scene (Fig. 2A), B) *INN* to learn the forward skinning and deformation of each element, enhancing the efficiency of the training process (Fig. 2B), C) *SDF module* to independently learn the geometry of each element (Fig. 2C) and D) *RGB Renderer module* to generate RGB values from individual SDF volumes for the final rendering (Fig. 2D).

**A. Semantic-aware ray sampling.** To reconstruct semantically separable geometry, it is essential that each 3D point in the reconstructed surface is tagged with a semantic label to denote its object affiliation. Previous efforts [5] propose to model a compositional scene within a shared network structure, which is challenging in the dynamic scenario for the following two reasons. First, it is hard to supervise the SDF of each entity from different frames by 2D semantic segmentation due to its dynamic nature. Second, the interaction between humans and objects in different frames is complex, considering the occlusion between each other. These issues provoke us to design a network structure with natural disentanglement for the compositional modeling.

To address these challenges, our network is designed with distinct modules for SDF prediction for each entity, maintaining semantic continuity from the input space to facilitate the prediction of semantically separable 3D geometry and joining them with the rendering module for composite rendering. Specifically, leveraging the 2D semantic mask of the input image, we sample two sets of rays surrounding the interacting objects. Along these rays, we sample 3D points to generate

two distinct sets: one representing the deformable object, $\{x_v^d\}_{i=1}^N$, and the other representing the non-deformable object, $\{x_v^{nd}\}_{i=1}^N$. However, only 2D semantic-aware ray sampling is not sufficient for accurate disentanglement of the individual entities, as the rays can intersect both entities in their path through the 3D space due to the occlusion of one object by the other. Hence, we also perform an encoding of the 3D points based on their distance from the 3D skeletons of individual entities. Please refer to Section 3C for more details about the semantic encoding of the 3D points. These points are then processed through separate networks tailored to each object type to get separate geometry and finally merged at the rendering module. This approach enhances our ability to effectively separate and reconstruct individual elements, even in scenarios of strong occlusions. This meticulous sampling strategy guarantees that supervision from individual entity's observation only influences each SDF prediction module.

**B. View space to canonical space deformation.** In dynamic NeRF, a common approach is mapping each frame to a fixed canonical frame to learn the correspondence between frames. For deformable objects, like humans, this is generally achieved by using pre-defined LBS weight from template models like SMPL *etc.* [7, 8, 39, 40]. LBS is a technique used to deform the surface points on a human mesh at the canonical pose to each frame's pose based on the positions and orientations of an underlying skeleton (Eqn. 1). A few methods, *e.g.*, TAVA [28] and SNARF [41] extend this approach beyond humans and learn the forward LBS for arbitrary deformable entities without using any template model. However, for learning the forward LBS, defined in canonical space, the main challenge lies in finding the correct correspondence in the canonical space for a given deformed point because of the implicitly defined correspondences without an analytical inverse form of the Eqn. 1. Hence, they formulate the problem of learning this correspondence as a root finding problem and solve for $\mathbf{x_c}$ *s.t.*, $LBS(\mathbf{w}(\mathbf{x_c}), \mathbf{B}, \mathbf{x_c}) - \mathbf{x_v} = 0$. They solve it numerically using iterative Broyden's method [43], which involves matrix-vector multiplications and computationally expensive matrix inversions at each iteration and must be optimized for multiple iterations for each iteration of NeRF optimization. Hence, this makes the training process time-consuming. To overcome this challenge, we replace this root finding problem and instead use an INN [29] for transforming the view space points to canonical space and learn the skinning weight $\mathbf{w}(\mathbf{x_c})$ simultaneously. Conditioned by each frame's pose, INN can correctly learn the bijective mapping between the deformed and canonical space through a consistency loss and bypassing the iterative optimization process, resulting in time-efficient training. Fig. 3 shows the workflow for the view to canonical space conversion framework.

First, function $\mathcal{G}_{\mathbf{v->c}}^{\mathbf{INN}}$ is used to transform the points from view space to canonical space using an INN defined as [29]. To enhance the convergence

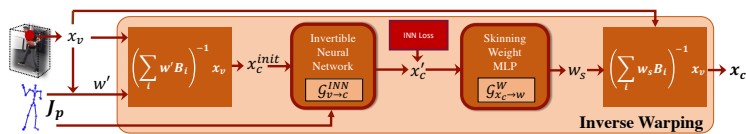

Figure 3: Overview of the transformation from view space to canonical space.

and provide a better initialization for the INN network, we first transform the deformed space points $\mathbf{x_v}$ to give an approximation of the canonical space points. Under the intuition that the sampled points $\mathbf{x_v}$ near the posed skeleton will remain close to the canonical skeleton at canonical space, $\mathbf{x_v}$ are transformed to $\mathbf{x_c^{init}} = (\sum_{i=1}^{n_b} \mathbf{w'}.\mathbf{B}_i)^{-1}.\mathbf{x_v}$, where, $\mathbf{w'}$ represents one-hot vectors defining the nearest joint of the posed 3D skeleton $\mathbf{J_P} \in \mathbb{R}^{n_b}$ from the view space points $\mathbf{x_v}$. An ablation study is presented in Tab. 6 to show the effectiveness of this initialization. The INN, $\mathcal{G}_{\mathbf{v->c}}^{\mathbf{INN}}$ is conditioned on per frame 3D skeleton $\mathbf{J_P}$. Then a skinning weight prediction network $\mathcal{G}_{\mathbf{x_c->w}}^{\mathbf{W}}$ is used to predict the skinning weight $\mathbf{w_s}$ at canonical space from the predicted canonical points $\mathbf{x_c'}$.

$$\mathbf{x_c'} = \mathcal{G}_{\mathbf{v->c}}^{\mathbf{INN}}(\mathbf{x_c^{init}}, \mathbf{J}) \qquad\qquad \mathbf{w_s} = \mathcal{G}_{\mathbf{x_c->w}}^{\mathbf{W}}(\mathbf{x_c'}) \qquad (2)$$

Finally, the canonical points are calculated as, $\mathbf{x_c} = (\sum_{i=1}^{n_b} \mathbf{w_s}.\mathbf{B}_i)^{-1}.\mathbf{x_v}$, that are given as input to the SDF prediction network. This helps to constrain the skinning weight prediction network. As skinning weight defines the weightage of underlying skeleton joints for the deformation of each vertex on the mesh surface; it helps capture better articulation or deformation and surface reconstruction under motion. Moreover, it implicitly learns the spatial relationship between the surface points, resulting in smoother reconstruction compared to the approaches where no skinning weight representation is learned. This is evident in Fig 5, given a comparison with such methods.

**C. Learning the SDF volume of individual elements.** Given our aim to generate semantically separable geometries of the scene entities, we use two independent SDF networks defined as $\mathcal{F}^j_{c->\Omega}(\theta_\Omega)$ for deformable and non-deformable objects $j \in \{d, nd\}$. The individual set of transformed canonical points $\mathbf{x_c}$ are passed through these geometry prediction networks, defined as $\mathcal{F}^j_{c->\Omega}(\theta) : \mathbb{R}^{3+3+n_b} \to \mathbb{R}^{1+256}$ (SDF and a global feature representation of dimension 256) produces surface reconstruction at canonical space, $\Omega^j_c$.

The SDF prediction network is conditioned on the canonical skeletons $\mathbf{J_{p0}} \in \mathbb{R}^{n_b}$. Only semantic-aware (based on a 2D semantic map) ray sampling is not sufficient to differentiate the points in 3D. For this purpose, to provide the SDF prediction network information about which object the point belongs to, we assign a semantic label to each 3D point. This is defined as a weightage, $\omega^j = \exp(-dist^2/\sigma^2)$ based on the distance of the point from the nearest 3D joints in the per entity canonical skeletons $\mathbf{J^j_{p0}}$. The points far from the canonical skeleton of either of the entities are assigned to the background, with weight defined as $\omega_{bg} = 1.0 - clamp(\sum_j \omega^j, 1.0)$. This semantic label with dimensionality 3 is concatenated with canonical point and pose input $(3 + 3 + n_b)$ and passed through the SDF prediction network that predicts SDF and a global feature representation.

**D. Compositing for final rendering.** We use a single RGB prediction network to predict the final rendering. For this purpose, the predicted canonical points (3), normals calculated from SDFs (3) [39], geometric features (256) and posed skeletons $\mathbf{J_{p0}}$ ($n_b$) for individual elements are concatenated before sending through a unified RGB prediction network $\mathcal{F}_{\Omega->rgb}$ Following Guo *et al.* [39] we predict the texture in canonical space and condition the texture generation network with normal calculated from SDFs to consider the deformation from view space points to canonical points. This is defined as $\mathcal{F}_{\Omega->rgb} : \mathbb{R}^{3+3+n_p+256} \to \mathbb{R}^3$.

**Training:** All the modules defined above are trained jointly over all the frames of the given video. The training losses used for the global optimization are defined as follows:

**- Reconstruction loss:** For optimizing the NeRF, reconstruction loss is defined between the rendered RGB $\hat{C}(r)$ and the RGB $C(r)$ of input pixel along the ray $r$, $\mathcal{L}_{rgb} = \sum_{r \in R} \|\hat{C}(r) - C(r)\|$.

Due to the lack of direct supervision of the skinning weight prediction network or INN, several incorrect combinations of canonical points and skinning weights can satisfy the forward LBS Eq. 1. Hence, the following supervisions are used on the skeleton space to constrain these two networks.

**- Pose loss:** To ensure that the INN network learns a correct mapping between the view space and the canonical space, a loss is defined on two sets of points $(X_{J_{p0}}, X_{J_p})$ sampled around the bones of canonical ($\mathbf{J_{p0}}$) and deformed skeletons ($\mathbf{J_p}$) respectively. The following loss is applied to ensure that the INN correctly transforms the point set $X_{J_p}$ to $X_{J_{p0}}$,

$$\mathcal{L}_{pose} = \sum_{p \in P} |X_{J_{p0}} - \mathcal{G}^{\mathbf{INN}}_{\mathbf{v}->\mathbf{c}}(X_{J_p})| \tag{3}$$

where $P$ is the total number of points sampled around the bones from each skeleton.

**- Skinning weight loss:** To constrain the skinning weight prediction network, a loss is applied to ensure that the predicted skinning weight for the canonical joints is a one-hot vector (1 for the respective joint and 0 for rests), $\hat{w}$.

$$\mathcal{L}_W = ||\mathcal{G}^{\mathbf{W}}_{x_c->\mathbf{w}}(J_{p0}) - \hat{w}||^2_2 \tag{4}$$

**- Cycle loss:** Conventional cycle consistency loss is used for optimizing the INN,

$$\mathcal{L}_{INN} = ||\mathcal{H}(\mathcal{H}^{-1}(\mathbf{x_v}, J_{p0}), J_p) - \mathbf{x_v}||^2_2 \tag{5}$$

where $\mathcal{H}$ is the transformation learned by the INN between view and canonical space. The inverse function $\mathcal{H}^{-1}(\mathbf{x_v}, J_{p0})$ transforms the view space point to canonical points, and the forward function $\mathcal{H}(\mathbf{x_c}, J_p)$ transforms back the canonical points to deformed space conditioned by the posed skeleton.

**- Consistency loss:** To ensure that the INN transformed point $\mathbf{x'_c}$ and the final skinning weight conditioned canonical points $\mathbf{x_c}$ (Fig. 3) are close to each other we also minimize the $L_2$ distance between these two sets of points.

$$\mathcal{L}_{Consis} = ||x'_c - x_c||^2_2 \tag{6}$$

**- In shape loss:** Following Guo *et al.* [39], to accelerate the learning process, a loss ($\mathcal{L}_{shape}$) is defined around a point cloud initialization for the individual entities. The point cloud is defined around the canonical skeleton of individual elements. This loss ensures that the transformed points $\mathbf{x_c}$ that fall within this point cloud have the sum of weights predicted from NeRF densities $\alpha = 1$

[39]. The full loss, minimized over each video frame, is defined as follows, where $\lambda_{skel}$, $\lambda_W$, $\lambda_{INN}$, $\mathcal{L}_{consis}$, $\lambda_{shape}$ are empirically set to 2,10,1,1,0.03 respectively.

$$\mathcal{L} = \mathcal{L}_{rgb} + \lambda_{skel}\mathcal{L}_{skel} + \lambda_W\mathcal{L}_W + \lambda_{INN}\mathcal{L}_{INN} + \mathcal{L}_{consis}\mathcal{L}_{consis} + \lambda_{shape}\mathcal{L}_{shape} \qquad (7)$$

**Evaluation:** As we learn the object geometry at canonical space, at inference time, we can directly sample points around the canonical skeleton and pass through the SDF prediction network $\mathcal{F}^j_{c->\Omega}(\theta)$ (Fig. 2C) to predict the canonical SDF. Then, for the canonical mesh vertices, we can predict the skinning weights using the skinning weight prediction network $\mathcal{G}^{\mathbf{W}}_{\mathbf{x_c}->\mathbf{w}}$ (Fig. 3) and generate the final posed mesh (viewing/deformed space) by following Eqn. 1.

**Implementation Details:** The overall network proposed in Fig 2 is trained end-to-end, with a learning rate of $5.0e-4$. We use ADAM optimizer for the SGD optimization and PyTorch library for the implementation of our method. All training and inference have been performed in NVIDIA RTX 4090 GPU. Details about the network architecture are presented in the Appendix.

# 4 Experiments

In this section, we first demonstrate the effectiveness of our method for precise 3D reconstructions of deformable and non-deformable objects, as well as their interactions. Next, we present qualitative and quantitative comparisons of our approach with relevant state-of-the-art methods. Additionally, we conduct a comprehensive ablation study to analyze the impact of different network design choices and loss formulations on our model's performance. Finally, we discuss the efficiency of our method in terms of training convergence time compared to existing approaches in the literature. **Datasets.** To evaluate our reconstruction under multiple entity interactions, we select the BEHAVE [31] with human-object interactions and HO3D-V3 [44] with hand-object interactions. We tested sequences featuring large motions of humans and objects, training with 45-50 images per camera view to optimize 3D geometry. Following [28], all methods utilize dataset-provided camera poses, body poses, and masks for training. We also evaluate our method for reconstructing single deformable entities (only human/animal) and use a similar setup as proposed in TAVA [28]. For this purpose, we test performance on two datasets: the ZJU-MoCap dataset [9] for human reconstruction and a synthetic dataset for animal reconstruction from [28].

**Baseline.** For human-object reconstruction, we choose methods that focus on generic scene reconstruction without using any template 3D models for a fair comparison of our template-free model. Specifically, we benchmark our approach against state-of-the-art generic scene reconstruction methods such as Tensor4D [25], NDR [15], HyperNeRF [27], and D-NeRF [14]. Additionally, we consider the recent approach ResFields, which models large and complex temporal motions by introducing temporal residual layers into the general NeRF architectures, showing significant improvements over the baselines [45]. We train NDR [15], HyperNeRF [27], and D-NeRF [14] with and without ResField layers on RGB videos. We haven't trained our method with ResField. For human reconstruction, we compare with methods that learn the skinning weight field [28, 27]. TAVA [28] learns the skinning weight from scratch without using any template model, whereas AnimatableNeRF [9] initializes the skinning weight from SMPL and learns a residual weight to optimize the final skinning weight field.

**Evaluation metrics.** For quantitative evaluation, following previous 3D reconstruction NeRFs [5, 6], we measure distance metrics (after registration between predicted and ground-truth meshes) such as Average Distance Accuracy (Dist. Acc.), Completeness, and Chamfer Distance (CD), along with Precision, Recall, and F-score (defined with a threshold of 5cm). Following previous methods [39], we use SMPL meshes provided in the ZJU-MoCap and BEHAVE datasets as ground-truth to evaluate human reconstruction quality to obtain the quantitative results. Similarly, we utilize the ground-truth object meshes within the BEHAVE dataset for object reconstruction evaluation.

**Results and discussions. - Reconstruction of scenes with two entities under interactions:** Our evaluation methodology for human-object reconstruction focuses on two critical dimensions: 1) *Holistic Scene Reconstruction*: We assess the entire scene's reconstructed mesh quality (human+object) against the ground truth, providing a comprehensive measure of the scene's accuracy (Tab. 3). 2) *Semantic Reconstruc-*

| Method \ Metric | Trained with ResField [45] | Dist. Acc. ↓ (cm) | Comp. ↓ (cm) | Prec. ↑ (%) | Recal. ↑ (%) | F-score ↑ (%) | Chamfer ↓ (cm) |
|---|---|---|---|---|---|---|---|
| Tensor4D [25] | ✗ | 4.152 | 2.441 | 69.413 | 91.642 | 78.993 | 3.297 |
| NDR [15] | ✗ | 4.203 | 3.527 | 73.048 | 78.846 | 75.599 | 3.865 |
| HyperNeRF [27] | ✗ | 4.125 | 3.362 | 73.510 | 80.683 | 76.661 | 3.742 |
| D-NeRF [14] | ✗ | 7.074 | 7.301 | 48.324 | 45.781 | 46.301 | 7.188 |
| NDR [15] | ✓ | 3.591 | 3.193 | 78.564 | 82.472 | 80.385 | 3.399 |
| HyperNeRF [27] | ✓ | 3.879 | 3.313 | 76.099 | 81.578 | 78.619 | 3.596 |
| D-NeRF [14] | ✓ | 3.927 | 3.364 | 75.774 | 81.453 | 78.398 | 3.646 |
| Ours | ✗ | **2.721** | **2.142** | **89.120** | **93.853** | **91.343** | **2.431** |

Table 3: Scene reconstruction results on BEHAVE [31] dataset.

| Method \ Metric | Trained with ResField [45] | Dist. Acc. ↓ (cm) | Comp. ↓ (cm) | Prec. ↑ (%) | Recal. ↑ (%) | F-score ↑ (%) | Chamfer ↓ (cm) |
|---|---|---|---|---|---|---|---|
| Tensor4D [25] | ✗ | 4.409 | 2.419 | 69.402 | 91.680 | 79.000 | 4.414 |
| NDR [15] | ✗ | 4.865 | 3.546 | 69.653 | 72.728 |  | 4.205 |
| HyperNeRF [27] | ✗ | 4.794 | 3.363 | 70.276 | 79.531 | 74.202 | 4.078 |
| D-NeRF [14] | ✗ | 6.515 | 6.132 | 53.013 | 52.745 | 52.071 | 6.324 |
| NDR [15] | ✓ | 4.681 | 3.186 | 72.934 | 81.025 | 76.553 | 3.931 |
| HyperNeRF [27] | ✓ | 4.265 | 3.394 | 73.864 | 79.642 | 76.331 | 3.831 |
| D-NeRF [14] | ✓ | 4.729 | 3.403 | 71.211 | 79.244 | 74.706 | 4.066 |
| Ours | ✗ | **1.761** | **1.863** | **97.225** | **93.624** | **95.343** | **1.812** |
| Tensor4D [25] | ✗ | 4.390 | 2.523 | 56.953 | 91.683 | 70.260 | 3.956 |
| NDR [15] | ✗ | 3.747 | 3.607 | 76.526 | 75.534 | 75.675 | 3.677 |
| HyperNeRF [27] | ✗ | 3.647 | 3.508 | 78.171 | 76.892 | 77.144 | 3.586 |
| D-NeRF [14] | ✗ | 4.675 | 5.529 | 64.88 | 54.095 | 57.748 | 5.102 |
| NDR [15] | ✓ | **3.442** | 3.531 | 81.114 | 76.612 | 78.641 | 3.485 |
| HyperNeRF [27] | ✓ | 3.451 | 3.282 | 80.551 | 80.720 | 80.174 | 3.379 |
| D-NeRF [14] | ✓ | 3.565 | 3.362 | 79.379 | 79.155 | 78.875 | 3.464 |
| Ours | ✗ | 3.571 | **2.121** | **82.762** | **92.410** | **86.991** | **2.741** |

Table 2: Semantic reconstruction results on BEHAVE [31] dataset. The upper and lower tables represent quantitative human and object reconstruction evaluation.

*tion*: We asses the reconstruction quality of individual scene elements by comparing the accuracy of reconstructed humans and objects to their respective ground truth meshes (Tab. 2). Our method excels in holistic scene reconstruction across all metrics (Tab. 3) and delivers superior results in semantic reconstruction, especially for humans (Tab. 2, upper rows) and most objects (Tab. 2, lower rows). Competing methods, focusing on the prior-free linkage between observation and canonical space, often neglect the topological relationships essential for dynamic scenes. While they are effective for rigid entities, they underperform in reconstructing high-quality, deformable entities. Our approach leverages learning skinning weights to capture the intricate relationships between body parts, enhancing reconstructions for both deformable and non-deformable objects (Fig. 5).

**-Reconstruction of scenes with hand-object reconstruction:** We evaluate our method also on the HO3D-V3 dataset [44] and present comparative results in Tab. 4. For this purpose, we choose two baseline methods, *i.e.*, NDR and HyperNeRF trained with Res-Field, that best perform on the BEHAVE dataset. Tab. 4 compares semantic reconstruction quality. We use the ground-truth mesh for hand from the HO3D-V3 dataset and the ground-truth mesh for objects from YCB-Video 3D models for calculating the metrics. Our method achieves better reconstruction, showing superior performance on most metrics. Also, we evaluate our method for single-object reconstruction, including arbitrary deformable entities,

**- Human reconstruction:** We compare our human surface reconstruction results

| Method \ Metric | Trained with with ResField [45] | Hand reconstruction | | | Object reconstruction | | |
|---|---|---|---|---|---|---|---|
| | | Dist. Acc. ↓ (cm) | F-score ↑ % | Chamfer ↓ (cm) | Dist. Acc. ↓ (cm) | F-score ↑ % | Chamfer ↓ (cm) |
| NDR [15] | ✓ | 1.419 | 94.051 | 1.217 | 1.154 | 93.782 | 1.279 |
| HyperNeRF [27] | ✓ | 1.435 | 93.491 | **1.198** | 1.159 | 97.988 | 1.042 |
| **Ours** | ✗ | **1.373** | **95.396** | 1.294 | **0.530** | **99.980** | **0.463** |

Table 4: Reconstruction results on HO3D-V3 dataset [44].

with TAVA [28] and AnimatableNeRF [7] on the ZJU-MoCap dataset [9] (Tab. 5, upper table). TAVA employs a template-free approach, while AnimatableNeRF uses the SMPL body model. Our method surpasses both TAVA and AnimatableNeRF in performance. The qualitative comparison, shown in Fig. 4, highlights the superior surface reconstruction of our method on the ZJU-MoCap dataset. SDF modeling contributes smoother surface reconstructions compared to models like [28, 7].

| Method \ Metric | Dist. ↓ Acc.(cm) | Comp. ↓ (cm) | Prec. ↑ (%) | Recal. ↑ (%) | F-score ↑ (%) | Chamfer ↓ (cm) |
|---|---|---|---|---|---|---|
| TAVA [28] | 2.79 | 2.14 | 90.40 | 95.67 | 92.95 | 2.47 |
| AnimatableNeRF [7] | 3.63 | 2.39 | 81.57 | 93.32 | 88.30 | 2.81 |
| Ours | **2.47** | **2.01** | **92.15** | **95.99** | **93.97** | **2.24** |
| TAVA [28] | 0.92 | **0.53** | 99.65 | 100.00 | 99.83 | 0.73 |
| Ours | **0.81** | 0.61 | **99.99** | **100.00** | **99.99** | **0.71** |

Table 5: Reconstruction on ZJU-Mocap (upper) [9, 46] and synthetic animal dataset [28] (lower).

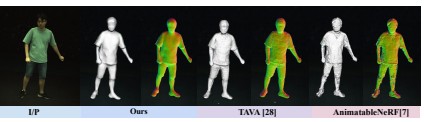

Figure 4: Qualitative comparison on ZJU-Mocap dataset [9].

**- Reconstruction of other deformable entities:** To evaluate our reconstruction method on other deformable entities, we employ a similar experimental setup as used in TAVA [28]. Tab. 5 (lower table) presents the quantitative comparison using the synthetic animal dataset introduced by TAVA. Our method shows comparable performance with the baseline method. Fig. 1 displays qualitative results from one of the animal subjects in the dataset.

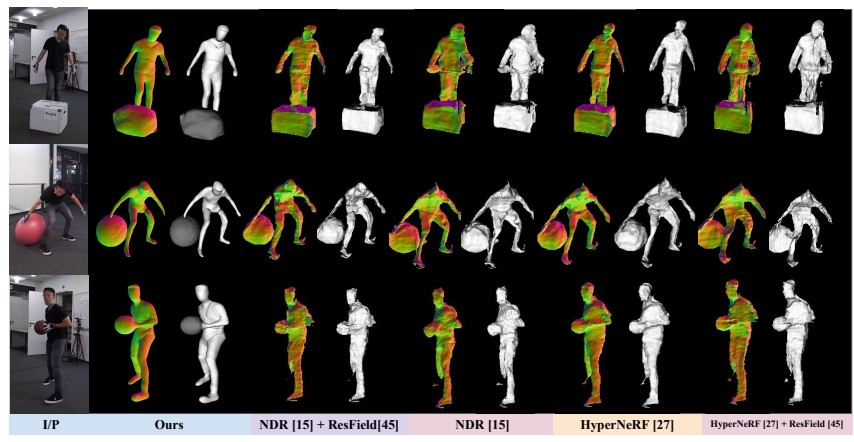

Figure 5: Qualitative comparison with SoTA methods on BEHAVE dataset.

**Ablation studies:** We present an ablation study for different network design choices and losses in Tab. 6. **Without initializing the INN with $x_c^{init}$:** In this experiment, we use an MLP to learn the initialization, following traditional INN networks [47], instead of initializing based on the distance of the deformed points from the posed skeleton (see Section 3, B.). Our initialization method leads to better reconstruction performance. **With the same geometry head:** Here, we use a unified network architecture with a single INN for both deformable and non-deformable objects, for predicting skinning weights and SDF. Rather than using semantic-aware ray sampling in image space for disentangling motions and geometries of entities (see Section 3, A), a semantic logit is predicted from the SDF and optimized with a semantic loss to produce semantically separable geometries following [5]. This approach struggles with high occlusion and complex interactions, resulting in poor reconstructions (Figure 6). **Without $\mathcal{L}_W$ and $\mathcal{L}_{pose}$ loss:** These losses are crucial for constraining the prediction networks. The $\mathcal{L}_W$ has a significant impact, effectively constraining the INN and skinning weight prediction, improving the reconstruction quality. **Training convergence for W and W/o Broyden method:** We experiment to assess the efficacy of our INN network compared to the Broyden-based LBS learning approach. For this purpose, we train our network to replace the INN network and use the Broyden equation solver to solve the Eqn. 1 and report the progress of CD with respect to time (Fig. 7). Our network, designed with INN, converges much faster than the Broyden method used in [28, 41]. Also, our approach takes around 0.3 sec/frame with INN compared to 0.9 sec/frame with the Broyden approach.

| Method \ Metric | Dist. ↓ Acc. (cm) | F-score ↑ (%) | Chamfer ↓ (%) |
|---|---|---|---|
| W/o $x_c^{init}$ (Section 3, B) | 3.92 | 83.50 | 3.51 |
| Same *geometry* heads (Section 3, A,B,C) | 7.83 | 41.18 | 8.42 |
| W/o $\mathcal{L}_W$ (Equ. 4) | 3.83 | 82.67 | 3.53 |
| W/o $\mathcal{L}_{pose}$ (Equ. 3) | 2.73 | 90.17 | 2.67 |
| **Ours** | **2.72** | **91.34** | **2.43** |

Table 6: Ablation on BEHAVE for holistic reconstruction.

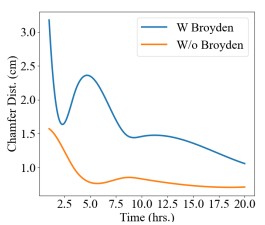

Table 7: INN vs Broyden formulation.

**Further study and limitations:** Although we successfully disentangle motions of the interacting entities, our framework currently employs separate networks for each entity, which is not scalable for scenarios involving more than two entities. This limitation affects its practicality for more com-

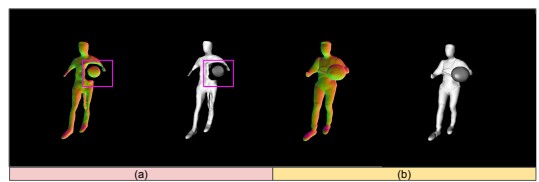

Figure 6: (a) Using single SDF network for both entities. (b) Using separate SDF networks for individual elements.

plex scenes with numerous interacting objects. A potential solution could involve the integration of an occlusion map to better manage interactions among multiple entities.

Also, we further analyze the dependency of reconstruction quality on 3D pose and semantic map accuracy (Table 8). *With predicted masks:* For this purpose, we have used a combination of a state-of-the-art object detection network, YOLOv8 [48], for first detecting the objects under reconstruction and then SAM [49] for segmenting the respective objects

| Evaluation Proc. | Type of Mask/Pose | Dist. Acc. ↓ (cm) | Comp. ↓ (cm) | Prec. ↑ (%) | Recal. ↑ (%) | F-score ↑ (%) | Chamfer ↓ (cm) |
|---|---|---|---|---|---|---|---|
| Human Recon. | GT | **1.761** | **1.863** | **97.225** | **93.624** | **95.343** | **1.812** |
| | Pred. mask | 2.002 | 2.248 | 96.003 | 90.619 | 93.233 | 2.125 |
| | Pred. mask + pose | 3.290 | 3.392 | 81.831 | 83.512 | 82.662 | 3.341 |
| Object Recon. | GT | **3.571** | **2.121** | **82.762** | **92.410** | **86.991** | **2.741** |
| | Pred. mask + pose | 3.694 | 2.408 | 81.152 | 91.600 | 86.060 | 2.901 |
| Scene Recon. | GT | **2.721** | **2.142** | **89.120** | **93.853** | **91.343** | **2.431** |
| | Pred. mask | 2.832 | 2.439 | 87.193 | 89.609 | 88.158 | 2.635 |
| | Pred. mask + pose | 3.283 | 3.143 | 80.765 | 85.364 | 83.001 | 3.213 |

Table 8: Reconstruction results on the BEHAVE dataset with predicted semantic masks and predicted pose.

within the predicted bounding boxes given by YOLOv8. Results show that our method can generate similar reconstruction results as dataset-given masks, because, in our method, the reconstruction quality for the semantically separable geometries is not solely dependent on the quality of input semantic masks. We utilize information from both 2D semantic masks and 3D skeletons. While the rays are sampled in image space within 2D bounding boxes around each entity, we also perform an encoding for every 3D point on the sampled rays based on its distance from the 3D skeletons of individual elements under reconstruction. *With predicted pose:* We generate these 3D skeletons by using the state-of-the-art 3D pose estimation network [50]. As the results show, our method needs a good quality 3D skeleton to constrain the shape and motions of the individual elements. Even though the reconstruction quality is not affected, the inaccuracies come from the predicted pose (Figure 7). So, this is another limitation of our method, that it requires good quality 3D pose for correct reconstruction.

# 5  Conclusion

This paper introduces TFS-NeRF, a 3D semantic NeRF framework for dynamic scene reconstruction using sparse/single-view RGB videos. Utilizing INN, our approach significantly streamlines the training process, addressing the typically long convergence times associated with existing template-free methods. Our method effectively disentangles the motions of multiple entities, whether

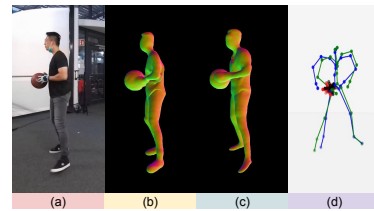

Figure 7: (a) Input image, (b), (c) Reconstruction with ground-truth and predicted pose, (d) pose inaccuracy.

rigid, non-rigid, or deformable, and optimizes per-entity skinning weights for accurate and semantically distinct 3D reconstructions. We conducted extensive experiments across various datasets, showcasing our approach's superior capability to manage complex interactions between multiple entities while ensuring high-quality reconstructions.

**Broader Impact.** Positive implications include advancements in digital media, robotics, and medical imaging, to name a few. Potential negative impacts involve privacy concerns and bias in model training, which can be mitigated by implementing strict data policies and ensuring diverse training datasets. Our proposal promises significant benefits for various fields, though it requires careful consideration of ethical and societal impacts.

**Acknowledgments.** This work has been partially funded by The Australian Research Council Discovery Project (ARC DP2020102427). We acknowledge the partial sponsorship of our research by the DARPA Assured Neuro Symbolic Learning and Reasoning (ANSR) program, under award number FA8750-23-2-1016.

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

## A    Appendix / supplemental material

### A.1    Neural Network architectures:

**Invertible Neural Network:** It transforms the view space points to canonical space (Fig. 3), $\mathbf{x_c'} = \mathcal{G}^{\mathbf{INN}}_{\mathbf{v->c}}(\mathbf{x_c^{init}}, \mathbf{J})$. We use realNVP [29] as the baseline of our Invertible Neural Network. This network consists of 2 Coupling layers [29], each with scaling and translation prediction modules. Each of these modules consists of 3 linear layers with dimensions $331 \times 512$, $512 \times 512$, and $512$. The input to the INN network is the deformed space points and it is conditioned on the skeleton pose. The input points are first transformed using a projection layer to dimension 256 which is concatenated with the skeletal pose (72) and passed as input to the scale and translation prediction networks. The predicted scale and translation transform the deformed space points to canonical space.

**Skinning weight prediction network:** The transformed canonical points are passed through the skinning weight prediction network $\mathbf{w_s} = \mathcal{G}^{\mathbf{W}}_{\mathbf{x_c->w}}(\mathbf{x_c'})$ (Fig. 3). The skinning weight prediction network consists of 3 linear layers with dimensions $3 \times 256$, $256 \times 256$, $256 \times 24$. The skinning weight prediction network takes the transformed canonical points as input, hence, the dimension is

3. The output defines the weightage of each skeleton joint on a 3D point for its deformation from canonical space to deformed space. Hence, the dimension of the output layer is $256 \times 24$. The output activation layer is defined as softmax as the sum of individual weight should be 1. Two similar architecture weight prediction networks are used for skinning weight prediction for individual entities.

**SDF prediction network:** Transformed canonical points are passed through the SDF prediction network Fig. 2 for geometry prediction, $\mathcal{F}^j_{c->\Omega}(\theta) : \mathbb{R}^{3+3+n_b} \to \mathbb{R}^{1+256}$. The SDF prediction network consists of 8 linear layers each with a hidden size of 256. A skip connection is added at layer 4. The dimensions of each layer are as follows $114 \times 256$, $256 \times 256$, $256 \times 256$, $256 \times 217$, $256 \times 256$, $256 \times 256$, $256$, $256 \times 257$. The input canonical points are transformed by a frequency layer and mapped to dimension 39, which is concatenated with canonical joints represented as $24 \times 3$ (each entity is represented by 24 skeleton joints). Moreover, as discussed in the main paper, each point is assigned a semantic label denoting which entity it belongs to *i.e.*, deformable, non-deformable object or background. The dimension of the semantic label is 3. Hence, the input dimension is $39 + 72 + 3 = 114$. The SDF prediction network generates an SDF value for each point (dimension 1) and a feature representation of dimension 256, resulting in a total output dimension of 257. We use separate networks for the SDF prediction of different entities. However, use similar architecture for both entities.

**RGB rendering network:** The RGB rendering network (Fig. 2) consists of 5 linear layers with dimensions of $270 \times 256$, $256$, $256$, $256 \times 256$, and $256 \times 3$. The input to the rendering network is canonical points with dimension 3, normals (calculated from predicted SDF) with dimension 3, and per-frame skeleton pose (concatenated skeletal joints from both the entities with dimension of $(72 \oplus 72)$ and the SDF predicted feature vector of dimension 256. A linear layer first transforms the skeleton pose to a lower dimension $(8)$. Hence, the input dimension of the rendering network is $3 + 3 + 8 + 256 = 270$. The output of the network is the RGB value for each sampled point, hence, the dimension of the output is 3. The output layer activation is defined as sigmoid.

