# OpenReview forum: "TFS-NeRF: Template-Free NeRF for Semantic 3D Reconstruction of Dynamic Scene"
_NeurIPS.cc/2024/Conference — NeurIPS 2024 poster_

### Official Review · Reviewer_UY9B · 2024-07-04

**Soundness:** 3
**Presentation:** 3
**Contribution:** 3
**Rating:** 5
**Confidence:** 3

**Summary:**

The author proposed a novel, time-efficient, template-free NeRF-based method for 3D dynamic scene reconstruction, focusing on capturing detailed explicit geometry for each entity in the scene. Extensive experiments demonstrate the efficiency of the proposed method.

**Strengths:**

1. The paper is well-organized, and the font size in the figures is appropriate, making it easy for readers to follow.
2. The method section is straightforward, providing a detailed introduction to each part, including sufficient details on the network, training, and loss weights.

**Weaknesses:**

1. The author mentioned that one of the contributions is time efficiency. Based on the description in lines 183-202, it seems the proposed method reduces computation to accelerate training. However, Figure 6 only shows the convergence speed. It would be more informative if the author could report the time consumed per iteration.

2. The information on how to obtain the semantic masks is missing. For example, in RoDynRF [1], the author uses optical flow and Mask R-CNN to obtain the segmentation masks.

3. The work focuses more on dynamic scenes with human subjects, but the title gives the impression that the method is designed for general dynamic scenes.

4. From Figures 2 and 3, it appears that the proposed method requires a 3D skeleton as part of the inputs. Does this mean that the proposed method does not work for scenes without humans or datasets lacking 3D skeleton information?

5. In the bottom part of Table 3, under the Dist. Acc column, HyperNeRF is the second-best method.

[1] Liu, Yu-Lun, et al. "Robust dynamic radiance fields." Proceedings of the IEEE/CVF Conference on Computer Vision and Pattern Recognition. 2023.

**Questions:**

See weaknesses.

**Limitations:**

The authors have addressed the limitations.

---

> ### Author Rebuttal · Authors · 2024-08-07
>
> **Time consumed per iteration:** Thank you for bringing this to our attention. Our approach takes around 0.3 sec/frame with INN while 0.9 sec/frame with the Broyden approach used in TAVA. We will include this information in our final paper.
>
> **Information on how we obtain semantic masks:** Please refer to the *Author Rebuttal* section for the response.
>
> **Gives impression of general dynamic scenes, but works on human subjects:** Please refer to the *Author Rebuttal* section for the response.
>
> **Method does not work for scenes without humans or datasets lacking 3D skeleton information:**
> We acknowledge that our method relies on 3D skeleton information to constrain the network and produce accurate shapes and geometries of the scene objects. However, this 3D skeleton information can be easily obtained for both rigid and non-rigid objects. For rigid objects, skeleton joints are derived from uniformly sampled 3D points along the x, y, and z axes, passing through the center of the 3D bounding box (obtained from multiview semantic masks) surrounding the object. For non-rigid objects, 3D skeletons can also be easily acquired using available tools (e.g. [3],[6], etc.).
>
> **Table 3, HyperNeRF is second best:** Thank you for pointing out the error. We will rectify the error in the final submission.
>
> [3] Yu Sun, Qian Bao, Wu Liu, Yili Fu, Black Michael J., and Tao Mei. Monocular, One-stage, Regression of Multiple 3D People. In ICCV, 2021. \
> [6] Alexander Mathis, Pranav Mamidanna, Kevin M. Cury, Taiga Abe, Venkatesh N. Murthy, Mackenzie W. Mathis, and Matthias Bethge. Deeplabcut: markerless pose estimation of user-defined body parts with deep learning. Nature Neuroscience, 2018.

---

> > ### Comment · Reviewer_UY9B · 2024-08-09
> >
> > Thank you for the detailed explanation. The authors have addressed all my questions, and I have decided to raise my score to borderline accept.

---

> > > ### Author Response · Authors · 2024-08-10
> > >
> > > Thank you for taking the time to review the rebuttal. We're glad to hear that our explanations were helpful and that you've decided to raise your score.

---

### Official Review · Reviewer_3yt5 · 2024-07-11

**Soundness:** 1
**Presentation:** 1
**Contribution:** 1
**Rating:** 3
**Confidence:** 4

**Summary:**

This paper propose TFS-NeRF to solve the semantic reconstruction of dynamic scenes.

**Strengths:**

Experimental results on multiple entity and deformable entity reconstructions are good.

**Weaknesses:**

It is very hard to follow the storyline of the introduction.

Unclear contributions. In Lines 81 and 91, the paper claims it could reconstruct multiple dynamic entities with multiple complex interactions, but in the experiment parts, the presented results are mostly from one person/animal with a single object, it is hard to judge whether the proposed model could process multiple dynamic entities with complex interactions. Also, the authors claim the proposed model could address the occlusion challenge in Line 81, but there is no experiments about it. Besides, the paper claims that it could reconstruct semantics of dynamic scenes by using semantic masks. However, such semantic reconstruction has been solved by previous works like Semantic Flow [1], it is hard to judge the contribution of this paper in this situation.

Missing important method details. This paper proposes a semantic-aware ray sampling strategy and illustrates the points are separated in to deformable object set and non-deformable object set, but does not present any details about the separation process. After reading the paper, the reviewer guesses it separates the points on the rays based on the semantic labels of the pixels traveling through the rays. However, it may not be correct since different points in the same camera ray may belong to different set (like a ray travels through a dynamic person and a static wall behind the person). It is very hard to understand the entire pipeline of semantic reconstruction without such details.

Writing should be significantly improved. There are many informal and unclear expressions in this paper:

- Line 405: Takeaways -> Conclusion
- Figure 2: Overview of the system-
- Line 229: unclear expression of R^{1+256}
- Line 221: why "it helps capture better articulation or deformation and surface reconstruction under motion?"
- Tables 5 and 6: what is the mearning of "->"
- Line 216: Given,
- Lines 98, 112, 136: the usage of : should be the same

Reference:
[1] Tian, Fengrui, Yueqi Duan, Angtian Wang, Jianfei Guo, and Shaoyi Du, Semantic Flow: Learning Semantic Fields of Dynamic Scenes from Monocular Videos, ICLR 2024.

**Questions:**

See above

**Limitations:**

Unclear expressions and limited contributions

---

> ### Author Rebuttal · Authors · 2024-08-06
>
> **Hard to follow introduction:** Thank you for your feedback regarding the introduction. We apologize if the storyline was challenging to follow. We will carefully review and revise the introduction to ensure it is clearer and more cohesive. Your input is valuable, and we appreciate your efforts to help us improve our work.
>
> **Claim on multiple dynamic entities:** Please refer to the *Author Rebuttal* section.
>
> **Claim on addressing occlusion challenge, but no experiment:**
> Thank you for the insightful review. We have included an experiment (**Figure 3 in the attached PDF in Author Rebuttal section**).
>
> Our goal is to generate a 3D reconstruction of the scene without using any template model. Unlike methods that use the T-pose or A-pose of SMPL models as canonical representations to handle self-occlusions, we select a keyframe from the sequence as the canonical pose. Consequently, if there is no frame in the sequence without occlusion, the SDF network struggles to capture the occluded geometry in the canonical space.
>
> To address this occlusion challenge, we use a strategy that disentangles the motions of distinct entities. This involves semantic-aware sampling and distance-based encoding of 3D points on the rays. Additionally, we employ separate SDF prediction networks for each entity, which helps mitigate the issue. Our overall design helps mitigate the occlusion problem.
>
> **Contribution compared to SemanticFlow** Thank you for mentioning this method. SemanticFlow primarily focuses on semantic rendering and scene editing. In contrast, our work is focused on semantic geometry reconstruction. Techniques that excel in rendering may not necessarily perform well in 3D geometry reconstruction ([4], [5], [7]). Therefore, our primary emphasis has been on methods that are strong in geometry reconstruction. However, we are considering evaluating this method to validate its effectiveness and to determine if it could be integrated into our final submission if appropriate.
>
> **Separates the points on the rays based on the semantic labels of the pixels** We apologize for not being clear in explaining this approach. We utilize information from both 2D semantic maps and 3D skeletons. While the rays are sampled in image space within 2D bounding boxes around each entity, we also perform an encoding for every 3D point on the sampled rays based on their distance from the 3D skeletons of individual elements. As mentioned in lines 230-238 of the initial submission for details, to provide the SDF prediction network information about which object the point belongs to, we assign a semantic label to each 3D point. This is defined as a weightage, $\omega^j = \exp(-dist^2/\sigma^2)$ based on the distance of the point from the nearest 3D joints in the per entity canonical skeletons $\mathbf{J^j_{p0}}$.
>
> **Formatting issues** Thank you for the detailed reviews and for pointing out these errors in the manuscript. We will rectify all these errors in the final paper.
>
> [4] Huang B, Yu Z, Chen A, Geiger A, Gao S. 2d gaussian splatting for geometrically accurate radiance fields. InACM SIGGRAPH 2024 Conference Papers 2024 Jul 13 (pp. 1-11). \
> [5] Guédon A, Lepetit V. Sugar: Surface-aligned gaussian splatting for efficient 3d mesh reconstruction and high-quality mesh rendering. InProceedings of the IEEE/CVF Conference on Computer Vision and Pattern Recognition 2024 (pp. 5354-5363).\
> [7] Martin-Brualla R, Radwan N, Sajjadi MS, Barron JT, Dosovitskiy A, Duckworth D. Nerf in the wild: Neural radiance fields for unconstrained photo collections. InProceedings of the IEEE/CVF conference on computer vision and pattern recognition 2021 (pp. 7210-7219).

---

> ### Author Response · Authors · 2024-08-13
>
> Dear Reviewer 3yt5,
>
> As the discussion period is drawing to a close, we would greatly appreciate it if you could let us know if there are any further clarifications or additional information we can provide to assist with any remaining questions.
>
> We have addressed the concerns you raised regarding unclear expressions for multiple dynamic entities, the lack of clarity on semantic-aware ray sampling, the perceived limited contribution compared to the Semantic Flow paper, and the missing experiments related to our claim on occlusion handling. Detailed responses to all these concerns are included in the **Author Rebuttal** and the review-specific rebuttal section. Additionally, we have added new experiments on occlusion handling in the attached PDF.
>
> Thank you once again for your thorough review and valuable suggestions on paper writing. We will ensure these points are addressed in the final version of the paper.

---

> > ### Comment · Reviewer_3yt5 · 2024-08-13
> >
> > Dear authors,
> >
> > Thank you for your effort in the rebuttal phrase. Your rebuttal has addressed some of my concerns.
> >
> > As for solving the occlusion problem is one of your contributions mentioned in the paper, although Figure 3 in the rebuttal seems to have good results, it is still weak to use only one sample to demonstrate the contribution.
> >
> > Although there is no further multiple dynamic entities dataset and the experiments are conducted on 2 entities, claiming the contribution on the generic scene may be somehow overqualified.
> >
> > Although Lines 230-238 illustrate the process of semantic label assignment, it is difficult to understand Lines 175-177 as it needs forward reference to get such information.
> >
> > Besides I am still confused about the "->" in Tables 5 and 6 and the unclear expression in Line 229.
> >
> > I feel more positive about this paper, but I still think the paper's contribution is not well-stated and the paper can not be accepted by the current version due to the writing issue.
> >
> > Thank you for your effort in the rebuttal again.

---

> ### Author Response · Authors · 2024-08-14
>
> Thank you for your feedback.
>
> - We would like to emphasize the main contributions of our paper:
>     - We present a time-efficient, template-free NeRF-based 3D reconstruction method for dynamic scenes involving two interacting entities.
>     - We focus on the semantic reconstruction of dynamic scenes, emphasizing the detailed and explicit geometry of each entity within the scene. \
>
>    While our system does address occlusion challenges that typically arise during entity interactions, it's important to note that handling occlusion is not the primary contribution of our work, and we did not claim it as such. Since this is not a main contribution, it should not be a reason for an unfavorable rating.
>
>  - We agree that "generic scene" or "multiple entities" terms can be misleading, and we are willing to revise these phrases. However, we believe these are minor adjustments.
>
>  - We agree that Lines 175-177 might be difficult to understand as they require forward reference for the details on label assignment. We believe this also needs only a minor revision, and we are happy to make these changes in the revised version.
>
>  - Regarding "$\downarrow$ Methods/Metric $\rightarrow$" in Tables 5 and 6: The $\rightarrow$ symbol indicates that the columns represent metrics. "$R^{1+256}$" defines that the SDF prediction network predicts both an SDF value and a global feature representation of the scene (as used in general NeRF). We will clarify these in the revised paper. These are straightforward fixes and are considered minor comments.

---

### Official Review · Reviewer_KrvP · 2024-07-13

**Soundness:** 3
**Presentation:** 2
**Contribution:** 3
**Rating:** 5
**Confidence:** 3

**Summary:**

The paper addresses the problem of reconstructing dynamic environments for arbitrary rigid, non-rigid, or deformable entities. The authors propose a template-free 3D semantic NeRF for dynamic scenes, which employs an Invertible Neural Network (INN) for LBS prediction, and optimizing per-entity skinning weights based on semantic masks. The experimental results show high-quality reconstructions of complex interactions with improved training efficiency compared to existing methods.

**Strengths:**

- The proposed method is able to reconstruct arbitrary non-rigid objects by utilizing TAVA framework.
- Disentanglement of objects improves the reconstruction quality by predicting LBS for each entity.
- The proposed framework outperforms existing template-free NeRF methods on various datasets containing different non-rigid and rigid objects.
- The paper is clearly written and easy to follow.

**Weaknesses:**

- The paper emphasizes semantic-aware ray sampling as a key contribution but lacks details on how the semantic masks are generated.
- The framework heavily relies on existing methods (TAVA, INN), limiting the novelty and distinctiveness of its contribution.

**Questions:**

- How is the semantic masks calculated? How is objects classified as rigid or non-rigid?
- During the training, are the same number of rays sampled for rigid and non-rigid objects?

**Limitations:**

- The method is not scalable for the scenes more than two entities.

---

> ### Author Rebuttal · Authors · 2024-08-06
>
> **How is the semantic masks calculated:** Please refer to the *Author Rebuttal* section.
>
> **Limiting novelty and distinctiveness w.r.t TAVA, INN:** Our method focuses on the research gap of producing semantic 3D reconstruction of dynamic scenes under multiple object interactions. While our problem setup of reconstructing the scene guided by a 3D skeleton is similar to TAVA, extending this concept for various object interactions is not trivial. Moreover, TAVA relies on an iterative method for solving the forward LBS, which is not time-efficient. In our method, we utilize INN to make the solution more time efficient, which makes it even more challenging under multiple object interactions setup, causing occlusion and different motions.
>
> **Classifier for rigid, non-rigid objects:** Thank you for your question. Based on the semantic masks, it is indeed possible to infer whether an object is rigid or non-rigid. We do not explicitly employ a separate classifier for this purpose. Instead, our approach leverages the information provided by the semantic masks to distinguish between rigid and non-rigid objects.
>
> **Number of rays sampled for rigid and non-rigid objects:** The rays are sampled in a 60:40 ratio for the non-rigid and rigid objects, respectively.

---

> > ### Comment · Reviewer_KrvP · 2024-08-12
> >
> > I appreciate the author's comments that helped clarify the paper. I initially did not assume that the authors used ground truth segmentation masks for the experiments. However, given that the method uses ground truth segmentation and 3D pose, it seems that the comparison in Tables 2, 3, and 4 is unfair, as the compared methods do not require segmentation masks or 3D skeleton information. Although the authors provided performance results using YOLO and SAM models in the rebuttal, the improvements over dynamic NeRF algorithms are marginal. Considering that the algorithm uses additional information (3D skeleton, skinning model, etc.), the experimental results diminish the contribution of the paper. Based on this and other reviewers' comments, I am downgrading my rating to borderline accept.

---

> ### Author Response · Authors · 2024-08-13
>
> Sorry if there is any confusion regarding the ground-truth segmentation masks.
>
> - All the methods (SOTA and ours) in Tables 2,3, and 4 in the initial submission are trained with ground-truth segmentation masks. So, the values for our method given in Table 1 of the attached PDF in rebuttal are not directly comparable with SOTA metrics in Tables 2,3, and 4 in the initial submission.
>
> - We acknowledge that we require 3D pose information which is not required by the SOTA methods. However, using a 3D skeleton helps in achieving much better reconstruction quality (as you will be able to find out in the qualitative results of Figure 5 in the main paper). Without using any constraints on the structure of the entities it is difficult to achieve a good reconstruction quality, especially when there are multiple entities present and entities are in large motions. Also, it should be noted that using the predicted poses does not affect the quality of the reconstruction (Figure 1 in rebuttal PDF) and the metric value decrease occurs due to predicted pose inaccuracies.
>
> -  We would also like to emphasize another point, that we do not use any off-the-shelf skinning model (like SMPL). Rather we predict the skinning weights (Figure 3 in the main paper) using an MLP and train an end-to-end network for the reconstruction.

---

> > ### Comment · Reviewer_KrvP · 2024-08-13
> >
> > Thank you for the clarification. I understand that the baseline methods also utilized ground truth masks. Regarding Tables 2 and 3, were the baseline NeRF methods trained separately on each entity (e.g., human and object), or was a single NeRF model used for the entire scene?

---

> ### Author Response · Authors · 2024-08-13
>
> Thank you for your question. Regarding Tables 2 and 3, a single NeRF model was trained for the entire scene, which includes both the human and the object.

---

> > ### Author Response · Authors · 2024-08-13
> >
> > Thank you. We are more than happy to clarify or address any additional questions you may have. If we are able to satisfactorily address the concerns that led to a lower rating, would you kindly consider revisiting the rating?

---

### Official Review · Reviewer_Rbt4 · 2024-07-17

**Soundness:** 3
**Presentation:** 2
**Contribution:** 3
**Rating:** 6
**Confidence:** 3

**Summary:**

This paper proposes TFS-NeRF, a semantic-NeRF framework leveraging no prior templates of dynamic scenes for 3D reconstruction. Guided by INN-driven LBS prediction and semantic-aware ray sampling, TFS-NeRF separately consider deformable and non-deformable parts during geometric learning but composite them to learn apperance to benefit from self-supervised RGB reconstruction loss. Experiments on human-object and animal cenric dynamic videos demosntrate the advantages of the proposed method against chosen baselibes.

**Strengths:**

-The overall paper is well motivated and easy to follow. The qualitative reuslts and supplement videos demonstrate the promising reconstruction performance of dynamic scenes with multiple entities.

-The experiment are extensively evalauted on several public benchmarks and the ablation studies highlight the unqiue contribution of several design choices.

**Weaknesses:**

My main concerns comes from the lack of clarification on several key components:

(1)As a semantic-nerf framework, how abou the influence of quality of input masks? As to labels from an imperfect 2D predictor, what is the impact on final results, since the label quality may affect the sampling quality for deformable and non-deformable rays.
(2)Similarly, what is the robustness to pose accuracy as poses are involved with both ray generation and the pose loss?

Discussion on the above two points could better strenthen the practicalibility of TFS-NeRF.

(3)As a rapidly growing area, the chosen baselines do not include latest methods such as HexPlan,Tensor4D or GS-based ones to highlight the advantages of using INNs, compositional rendering as well as reconstruction quality.

(4)As to LBS predicton, what is the unique advantages of applying INN over other types of NN modules (e.g, lossy CNN auto-encoders)? What is the efficiency or performance boosts compared to standard CNNs?

(5)How to get the initial values of B_i to compute the initialization value of x_c for arbitrary objects?

(6)As the paper is claimed to template free and able to deal with generic scenes, it would be more exciting and convincing to see other types of interactions? For example, a common daily objects (e.g. a door or a cabinet, opening and closing) to really show the advantages of prior-free deformations of general scenes.

(7)It would be good to aldo add the appearance quality or view synthesis performanc, though I understand the focus of TFS-NeRF is on 3D reconstruction.

(8)Formatting needs to be improved. For example, it is wierd that in Sec3.B, authors put much related work descriptions. There are also improper indents like Line 282 and Line284,etc.

**Questions:**

Please see the weakness section above on the clarification on the robustness to masks and pose qualities, as well as more in-depth discussions towards recent baselines and more general scenes.

**Limitations:**

Limitations are properly mentioned in the main paper.

---

> ### Author Rebuttal · Authors · 2024-08-06
>
> **Robustness to mask accuracy:** Please refer to the *Author Rebuttal* section.
>
> **Robustness to pose accuracy:** We agree that for the practicability of TFS-NeRF, it is important to evaluate it from this aspect, and apologize for not presenting the same in the initial submission. We have added quantitative results for the reconstruction quality of our method using predicted 3D poses (**Table 1 in the attached pdf in Author Rebuttal section**). We generate these 3D poses by using the state-of-the-art 3D pose estimation network [3].  As the results show, our method needs a good quality 3D pose to constrain the shape and motions of the individual elements. Even though the reconstruction quality does not get affected, the inaccuracies come from the predicted pose (**Figure 1 in the attached pdf in Author Rebuttal section**). We will include this evaluation in the final paper.
>
> **Comparison with current methods (Hexplane, Tensor4D, GS-based methods):** Thank you for highlighting these methods. We primarily compared our approach with techniques that emphasize geometry reconstruction, such as NDR, ResField, D-NeRF, and HyperNeRF, which predict Signed Distance Functions (SDF) for underlying geometries. We agree that including Tensor4D in the comparison is valuable and have conducted additional experiments to assess its performance relative to our method (**Figure 2 and Table 2 in attached pdf in Author Rebuttal section**).
>
> Our experiments show that our method provides superior reconstruction quality compared to Tensor4D. While Tensor4D's key innovation lies in its efficient 4D tensor decomposition, which represents dynamic scenes as a 4D spatiotemporal tensor, it does not account for the relative motions between scene elements. Instead, it captures the scene's overall motion. In contrast, our approach models the motion of individual elements and employs skeleton-guided reconstruction, leading to more accurate geometry.
>
> Additionally, current GS-based methods for dynamic scenes primarily focus on rendering rather than surface reconstruction. Research indicates that methods emphasizing geometry reconstruction often fall short in rendering quality, and vice versa (refer to Table 3 in 2DGS [4], Table 1 in SuGaR [5]). Similarly, Hexplane prioritizes image rendering over geometry reconstruction which is a NeRF-based method. Similar arguments are applicable for the NeRf-based method [7]. Therefore, these methods may not be directly comparable. Hence, given the limited time frame for rebuttal, we are concentrating on the geometry-based method. However, we are open to including these methods in the final comparison.
>
> **INN vs Lossy auto-encoder:** We appreciate your feedback and would like to clarify any confusion. We chose INN because we expect to maintain the invertible mapping between the 3D point space and the deformation space, which is the property that INN naturally has. A lossy auto-encoder could also become a choice only if it has the same property. We left the model structure exploration as future work to enhance the semantic reconstruction task further.
>
> **Computation of $B_i$ for arbitrary objects:** We apologize for the lack of clarity in the submission. For arbitrary non-articulated objects, $B_i$​ is computed based on the 3D bounding boxes, which are derived from multiview semantic masks. Specifically, for non-articulated objects, $B_i$​ represents the rigid transformation between the bounding boxes of the canonical frame (keyframe) and each input frame. We will ensure that this explanation is included in the final version of the paper to provide clearer context.
>
> **Evaluation on more datasets (daily life objects with door close, opening):** Thank you for highlighting this important aspect. Given the limited time frame for rebuttal, we are unable to evaluate additional datasets at this moment. Moreover, we have already evaluated several benchmark datasets (including datasets like BEHAVE containing daily life objects) to showcase the efficacy of our method. We appreciate your understanding and interest in expanding the scope of our evaluation.
>
> **Rendering quality:** We appreciate your feedback regarding the importance of appearance quality. We have compared our rendering quality with Tensor4D and obtained the following results: PSNR of 30.09 and SSIM of 0.970, whereas Tensor4D achieved a PSNR of 35.78 and an SSIM of 0.985. Methods, that show very good geometric reconstruction quality do not necessarily perform well in rendering and vice versa ([4], [5], [7]). The improvements in geometric detail often reduce the rendering quality, as the optimization process becomes more complex and computationally demanding.​ In our work, we aim to do good reconstruction but we will not care about rendering quality.
>
> **Formatting issues:** Thank you for the detailed reviews and feedback. We will rectify all these errors in the final paper
>
> [3] Yu Sun, Qian Bao, Wu Liu, Yili Fu, Black Michael J., and Tao Mei. Monocular, One-stage, Regression of Multiple 3D People. In ICCV, 2021.\
> [4] Huang B, Yu Z, Chen A, Geiger A, Gao S. 2d gaussian splatting for geometrically accurate radiance fields. InACM SIGGRAPH 2024 Conference Papers 2024 Jul 13 (pp. 1-11).\
> [5] Guédon A, Lepetit V. Sugar: Surface-aligned gaussian splatting for efficient 3d mesh reconstruction and high-quality mesh rendering. InProceedings of the IEEE/CVF Conference on Computer Vision and Pattern Recognition 2024 (pp. 5354-5363).\
> [7] Martin-Brualla R, Radwan N, Sajjadi MS, Barron JT, Dosovitskiy A, Duckworth D. Nerf in the wild: Neural radiance fields for unconstrained photo collections. InProceedings of the IEEE/CVF conference on computer vision and pattern recognition 2021 (pp. 7210-7219).

---

> > ### Author Response · Authors · 2024-08-13
> >
> > Dear Reviewer Rbt4,
> >
> > As we approach the end of the discussion period, we would like to know if there are any further clarifications or additional information we can provide to address your concerns.
> >
> > Currently, to address the issue of robustness to input masks and pose accuracy, we have conducted additional experiments. The results are presented in **Table 1 and Figure 1 of the attached PDF**. These experiments analyze the quality of the reconstruction in relation to these factors.
> >
> > Regarding the concern about comparisons with recent baselines, given the limited time for the rebuttal, we have focused on the most relevant baselines that specifically target 3D reconstruction. These comparisons are presented in **Table 2 and Figure 2 of the attached PDF**. We also provide a discussion in the **rebuttal section** below,  explaining why the other baselines are not directly comparable.
> >
> > Also, we have included all the details (in **rebuttal section**) about the method mentioned in the Weaknesses section.
> >
> > Thank you once again for your time and would be glad to address any further queries.

---

> > ### Comment · Reviewer_Rbt4 · 2024-08-13
> >
> > I thank authors for providing a detailed response to most of my concerns during the rebuttal period. And I have read all comments from all reviewers carefully.
> >
> > The ablative stuides on the quality of input priors are indeed important to this paper as one main advantages of this submisison is to get rid of explicit templates like SMPL. I would encourage authors to make a more extensive studies on  various scenes to help readers know the performance bound of this paper.
> >
> > The comparison to Tensor4D interms of NVS and reconstructions are valuable and reveals performance gaps and tradeoffs. I would expect authors to give further motivations for INN qualitatively and quantitatively as well.
> >
> > Overall, though there are still monir points to be improved in the revision, I think the current form provide necessary information and show the potential of practical usages given noisy priors. I would like to raise to my score to a positive rating.

---

> > > ### Author Response · Authors · 2024-08-13
> > >
> > > Thank you very much for your thoughtful feedback and for taking the time to carefully review our responses and the comments from all reviewers, as well as for raising your score. We will certainly take your suggestions to conduct more extensive studies across various scenes and perform additional experiments to clarify the motivations for using INN to strengthen these aspects in our revision.

---

### Author Rebuttal · Authors · 2024-08-06

We are grateful for the constructive feedback from the reviewers and are pleased that they found our paper "well-motivated, well-organized, and easy to follow" (Rbt4, KrvP, UY9B). They highlighted that our paper includes a "detailed introduction, sufficient details, and extensive evaluation on several public benchmarks" (UY9B, Rbt4). Additionally, they noted that our method produces "promising reconstruction under multiple entities" (Rbt4, 3yt5). We appreciate these insights and will incorporate all the suggestions and additional experiments to enhance the writing and presentation of the paper. In this section, we try to address all the common concerns.

**How masks are generated and the influence of the quality of masks on results [Rbt4, UY9B]:** We appreciate for highlighting this point. Following the baseline method TAVA,  we use the semantic masks given in respective datasets, to generate the quantitative results for ours as well as the comparing methods. \

As per the suggestion, to evaluate the robustness of our method in terms of the accuracy of semantic masks, we have further provided an experiment with predicted semantic masks (**Table 1 in attached pdf**). For this purpose, we have used a combination of a state-of-the-art object detection network, YOLOv8 [1], for first detecting the objects under reconstruction and then SAM [2] for segmenting the respective objects within the predicted bounding boxes given by YOLOv8. Results show that our method can generate similar reconstruction results as *dataset given masks*, because, in our method, the reconstruction quality for the semantically separable geometries is not solely dependent on the quality of input semantic masks. We utilize information from both 2D semantic masks and 3D skeletons. While the rays are sampled in image space within 2D bounding boxes around each entity, we also perform an encoding for every 3D point on the sampled rays based on its distance from the 3D skeletons of individual elements under reconstruction (Please refer to lines 230-238 of the initial submission for details). Hence, our method does not need a very accurate semantic mask as input. We will include this evaluation in the final paper.

**Evaluation dataset (mentioned generic scene but results given only on two entities, mostly focused on the human, or a single entity) [3yt5, UY9B]** We apologize for the lack of clarity and confusion in our presentation. By *multiple dynamic entities* in our paper, we mean more than one entity interacting with each other. Our method considers only two entities as of now. We have discussed this limitation under the *limitations and future directions* section (Lines 398-404 in the submission). The results on single entities (only human or animal subjects) are presented to demonstrate the capability of our reconstruction methods for *arbitrary deformable subjects* without using any template.\
For datasets with more than one entity, we selected two types: 1) human-object and 2) hand-object interaction datasets. To the best of our knowledge, there is no available dataset where an animal interacts with any object.\
We agree with the reviewer that the phrase multiple dynamic entities could be misleading, suggesting that our method can handle more than two entities. We will revise this phrase in the final paper to ensure clarity.

**[1]** Glenn Jocher, Ayush Chaurasia, and Jing Qiu. Ultralytics YOLO, January 2023.\
**[2]** Alexander Kirillov, Eric Mintun, Nikhila Ravi, Hanzi Mao, Chloe Rolland, Laura Gustafson, Tete Xiao,
Spencer Whitehead, Alexander C Berg, Wan-Yen Lo, et al. Segment anything. In Proceedings of the
IEEE/CVF International Conference on Computer Vision, pages 4015–4026, 2023.

---

> ### Author Response · Authors · 2024-08-10
> **Regarding further queries**
>
> **Dear Reviewer Rbt4, KrvP, 3yt5**
>
> Thank you once again for your thoughtful reviews. Your insights have significantly helped in improving the quality and clarity of our paper. We sincerely hope that our rebuttal has adequately addressed your questions and concerns. Please let us know if there are any further clarifications or additional information you require. We appreciate your valuable suggestions.
>
> Thank you very much for your time.

---

### Decision · Program_Chairs · 2024-09-25

**Decision:**

Accept (poster)

**Comment:**

The authors and reviewers engaged in a detailed discussion. Thank you all for your time!
After the discussion most concerns could be addressed by additional explanation and results. Some points still remain open and the score average borderline, with one weak reject. All reviewers provided detailed reviews, with two of them increasing their initial rating (Rbt4 from 4 to 6 and UY9B 4 to 5). The one negative review remains critical about the motivation, clarity, and overall writing. The AC agrees that these points could be improved but points are relatively minor, with the setting (two objects interacting) already being well explained. Hence, the paper can be accepted with minor revisions to the final version concerning clarity. Please incorporate the issues raised by 3yt5 carefully.